# Area law of noncritical ground states in 1D long-range interacting systems

Tomotaka Kuwahara [1,2 ✉] & Keiji Saito[3]

The area law for entanglement provides one of the most important connections between information theory and quantum many-body physics. It is not only related to the universality of quantum phases, but also to efficient numerical simulations in the ground state. Various numerical observations have led to a strong belief that the area law is true for every non-critical phase in short-range interacting systems. However, the area law for long-range interacting systems is still elusive, as the long-range interaction results in correlation patterns similar to those in critical phases. Here, we show that for generic non-critical one-dimensional ground states with locally bounded Hamiltonians, the area law robustly holds without any corrections, even under long-range interactions. Our result guarantees an efficient description of ground states by the matrix-product state in experimentally relevant long-range systems, which justifies the density-matrix renormalization algorithm.

[1] Mathematical Science Team, RIKEN Center for Advanced Intelligence Project (AIP), 1-4-1 Nihonbashi, Chuo-Ku, Tokyo 103-0027, Japan. [2] Interdisciplinary Theoretical & Mathematical Sciences Program (iTHEMS) RIKEN 2-1, Hirosawa, Wako, Saitama 351-0198, Japan. [3] Department of Physics, Keio University, Yokohama 223-8522, Japan. ✉email: tomotaka.kuwahara@riken.jp

The quantum entanglement plays a crucial role in characterizing the low-temperature physics of quantum many-body systems in terms of quantum information science. It is often measured by the quantum entanglement entropy between two subsystems, and its scaling is deeply related to the universality of the ground state[1,2]. When the interactions in quantum many-body systems are local, the quantum correlation is typically expected to be short range. This intuition leads to the conjecture that the entanglement entropy naturally scales as the boundary area of the subregion. This *area-law conjecture* is numerically verified in various quantum many-body systems, and is expected to be true in all gapped ground states (i.e., in non-critical phases)[3].

In one-dimensional (1D) systems, for an arbitrary decomposition of the total system, the area law for a ground state is simply stated as follows (Fig. 1):

$$S(\rho_L) \leq \text{const.}, \quad \rho_L = \text{tr}_R(|0\rangle\langle0|), \quad (1)$$

where the ground state is denoted as $|0\rangle$ and $S(\rho_L)$ is the von Neumann entropy, namely $S(\rho_L) = \text{tr}(-\rho_L \log \rho_L)$. Over the past dozen years or so, the area-law conjecture has attracted much attention, as it characterizes the universal structure of many-body physics in simple and beautiful ways[3]. However, providing detailed proof of the area law is still an extremely challenging problem. So far, the proof of this law is limited to gapped 1D systems[4–7], 1D quantum states with finite correlation lengths[8,9], gapped harmonic lattice systems[10,11], tree-graph systems[12], and high-dimensional systems with specific assumptions[13–17] (see ref. [3] for a comprehensive review). The area law is the backbone of the density-matrix renormalization algorithm[18], as it implicitly assumes the area-law structure of the ground states. The results pertaining to the 1D area law[4,6] rigorously justify the efficient description of the ground states using the matrix-product state (MPS), which facilitates the calculation of the ground states by the classical polynomial-time algorithm[7,19]. Finally, in the characterization of ground states, complete classification of 1D quantum phases has been achieved under the MPS ansatz[20].

Recent experimental advances have enabled the fine-tuning of the interactions between particles[21–24]. These advances push the long-range interacting systems from the theoretical playground to the field relevant to practical applications. One of the examples of controllable 1D long-range interacting spin systems is the following long-range transverse Ising model:

$$H = \sum_{i<j} \frac{J_{i,j}}{r_{i,j}^{\alpha}} \sigma_i^x \sigma_j^x + B \sum_i \sigma_i^z, \quad (2)$$

with $\{\sigma^x, \sigma^y, \sigma^z\}$ as the Pauli matrices, where $r_{i,j}$ is the distance between the two sites $i$ and $j$, and the exponent is tunable from $\alpha = 0$ to $\alpha = 3$[22,24] (also $\alpha = 6$ by van-der-Waals interactions[25,26]). In theoretical studies, new types of quantum phases induced by long-range interactions have been reported in the transverse Ising model[27,28], the Kitaev chain[29,30], the XXZ model[31], the Heisenberg model[32,33], as well as other models.

Typically, nontrivial quantum phases are induced by long-range interactions with power exponents smaller than three ($\alpha \leq 3$). For $\alpha > 3$, the universality class is the same as that of short-range interacting systems[34,35] (i.e., $\alpha = \infty$). This means that the regime of $\alpha \leq 3$ is essentially important to the discussion of the area law in long-range interacting systems.

We can now turn to the question of whether the area law of the entanglement entropy (1) is still satisfied in the presence of long-range interactions. Typically, long-range interacting systems show a power-law decay of the correlations even in noncritical ground states[27,29]; this property is similar to critical ground states in short-range interacting systems. To date, it has been a challenge, both numerically and theoretically, to identify the regime of $\alpha$ to justify the area law. Although several numerical studies suggest that the area law holds for short-range regimes (i.e., $\alpha > 3$), the possibility of a sublogarithmic violation to the standard area law (1) has also been indicated for $\alpha \leq 3$[27]. On the other hand, most theoretical analyses regarding the area law rely on the strict locality of the interactions, and cannot be directly applied to the power-law decay of interactions even for sufficiently large $\alpha$ values.

One of the natural routes to prove the area law under long-range interactions is to connect the entanglement entropy to the power-law decay of the bipartite correlation by extending the area-law proof from exponential clustering[8,9]. However, such a connection cannot be generalized because of the existence of strange quantum states[36] that have arbitrarily large entanglement entropy values while maintaining a correlation length of order $\mathcal{O}[\log(n)]$ (i.e., corresponding to $\alpha = \infty$). The other route relies on assuming the existence of the quasi-adiabatic path[37] to a trivial ground state satisfying the area law. Using the small-incremental-entangling theorem, this assumption ensures the area law in generic gapped short-range interacting systems[38]. However, regarding 1D long-range interacting systems, the area law has been proved only for short-range regimes $\alpha > 4$ even under this strong assumption[39].

Based on the above discussion, we report a general theorem on the area law in 1D long-range interacting systems in this work. It applies to generic 1D gapped systems with $\alpha > 2$ and ensures a constant-bounded entanglement entropy even in long-range regimes ($\alpha \leq 3$) in which nontrivial quantum phases appear owing to their long-range nature. We provide an outline of the proof in the "Methods" section.

## Results

**Main statement on the area law.** We consider a 1D system with $n$ sites, each of which has a $d$-dimensional Hilbert space. We focus on the Hamiltonian $H$ with power-law decaying interactions

$$H = \sum_{i<j} h_{i,j} + \sum_{i=1}^{n} h_i, \quad (3)$$

with $\|h_{i,j}\| \leq J/r_{i,j}^{\alpha}$ and $\|h_i\| \leq B$ for $\forall i, j$, where $\{h_{i,j}\}_{i<j}$ are the bipartite interaction operators, $\{h_i\}_{i=1}^{n}$ are the local potentials, and $J$ and $B$ are constants of $\mathcal{O}(1)$. One typical example is given by the long-range Ising model, shown in Eq. (2), where $d = 2$, $h_{i,j} = J_{i,j}\sigma_i^x\sigma_j^x/r_{i,j}^{\alpha}$ and $h_i = B\sigma_i^z$. As long as the local energy is finitely bounded, our result can also be extended to fermionic and bosonic systems (e.g., hard-core bosons). For simplicity, we here restrict ourselves to two-body interactions, but our results are generalized to generic $k$-body interactions with $k = \mathcal{O}(1)$. We consider the entanglement entropy of the ground state $|0\rangle$ in terms of the spectral gap $\Delta$ just above the ground-state energy. We assume that the ground state is not degenerate.

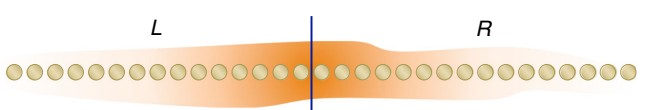

$L$ $R$

**Fig. 1 Area law in 1D system.** When we decompose the total system into two subsystems $L$ and $R$, the boundary area between the two subsystems is described by points. The area law simply argues that the entanglement entropy is bounded from above by a constant value, as in Eq. (1). We investigate the robustness of the area law under long-range interactions, which induce nonlocal quantum correlations.

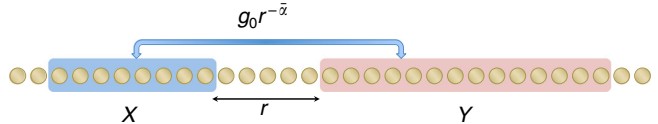

**Fig. 2 Condition for the area law in long-range interactions.** For arbitrary subsystems $X$ and $Y$ separated by $r$ from each other, we assume the total interaction strength between $X$ and $Y$ decay as $r^{-\bar{\alpha}}$, as shown in (5). This condition implies that $\alpha > 2$ in Eq. (3) if we consider the most general class of long-range interacting systems. On the other hand, if we restrict ourselves to a special class of fermionic systems with long-range hopping (7), the condition is relaxed to $\alpha > 3/2$.

We now discuss our main theorem. We define the interaction between two concatenated subsystems $X$ and $Y$ as follows (Fig. 2):

$$V_{X,Y} = \sum_{i \in X} \sum_{j \in Y} h_{i,j}. \qquad (4)$$

It simply selects all the interaction terms $\{h_{i,j}\}_{i<j}$ between two sites in $X$ and $Y$. Here, we assume the existence of a constant $g_0 \geq 1$ such that

$$\| V_{X,Y} \| \leq g_0 r^{-\bar{\alpha}} \quad (\bar{\alpha} > 0), \qquad (5)$$

for arbitrary choices of $X$ and $Y$, separated by a distance $r$. Under condition (5), the entanglement entropy $S(\rho_L)$ is bounded from above by

$$S(\rho_L) \leq c \log^2(d) \mathcal{G}\left(\frac{\log(d)}{\Delta}\right), \qquad (6)$$

for arbitrary choices of $L$ and $R$, where $\mathcal{G}(x) := x^{1+2/\bar{\alpha}} \log^{3+3/\bar{\alpha}}(x)$ and $c$ is a constant that depends on $\alpha$, $J$, $B$, $\bar{\alpha}$, and $g_0$. When the local dimension $d$ and the spectral gap $\Delta$ are independent of the system size $n$, the above inequality results in a constant upper bound for the entanglement entropy. Our area-law result can also be applied to quasi-1D systems (e.g., ladder systems) by appropriately choosing the local dimension $d$.

**Why does the area law hold for $\alpha > 2$?** We here show a physical intuition behind our long area law (6). Naively, the area law might be derived from the power-law decay of the bipartite correlations[40]. However, this behavior of the correlation functions is also observed in critical ground states, where the area law is usually known to be violated[1,2]. Moreover, as has been mentioned, the entanglement entropy can obey the volume law for a quantum state with super-polynomially decaying correlations[36]. At first glance, these points are contradictory to our results. In order to resolve this, we need to focus on the fact that the gap condition imposes much stronger restrictions on the entanglement structure of the ground states than the decay of bipartite correlations (see refs. [41,42] for example). Our proof approach fully utilizes the gap condition. This point is reflected to the approximation of the ground state using a polynomial of the Hamiltonian, where the approximation error increases as the spectral gap shrinks (see Claim 3 in the "Methods" section).

We also mention why the condition $\alpha > 2$ is a natural condition for the long-range area law. If the exponent $\alpha$ is small enough such that condition (5) breaks down, the norm of the boundary interaction along a cut (i.e., $V_{X,Y}$ with $X = L$ and $Y = R$) diverges in the thermodynamic limit ($n \to \infty$). Then, the system energy possesses a high-dimensional character, and hence its 1D character should be lost.

In order to study this point in more detail, let us consider the area law for thermal equilibrium states, namely $\rho = e^{-\beta H}/\mathrm{tr}(e^{-\beta H})$. A natural extension of the ground state's area law is to consider the mutual information $I_\rho(L:R) := S(\rho_L) + S(\rho_R) - S(\rho)$. Note that the mutual information is equal to the entanglement entropy in the limit of $\beta \to \infty$. At arbitrary temperatures, ref. [43] has provided the upper bound of $I_\rho(L:R) \leq 2\beta \|V_{L,R}\|$ (see also ref. [44]), which becomes a constant upper bound (i.e., the area law) if $\|V_{L,R}\| = \mathcal{O}(1)$. On the other hand, the area law may collapse for $\alpha \leq 2$, where the norm of $V_{L,R}$ can diverge to infinity in the thermodynamic limit. It is natural to expect that the condition for the area law in the thermal state should be looser than that in the ground state. This intuition indicates that the condition of $\alpha > 2$ should be, at least, a necessary condition for the area law of the entanglement entropy in the ground state. We have actually proved that $\alpha > 2$ is the sufficient condition. We thus believe that our condition of $\alpha > 2$ is already optimal (see also the "Discussion" section below).

**Several remarks on the area law.** There are several remarks pertaining to the above area-law results. First, in the short-range limit (i.e., $\bar{\alpha} \to \infty$), our area-law bound reduces to the following upper bound:

$$S(\rho_L) \leq c \frac{\log^3(d)}{\Delta} \log^3\left(\frac{\log(d)}{\Delta}\right) \quad \text{for} \quad \bar{\alpha} \to \infty.$$

This upper bound reproduces the state-of-the-art bound in short-range interacting systems[6,7]. This implies that our result provides a natural generalization from the short- to the long-range area law.

Second, assumption (5) is always satisfied for $\alpha > 2$ because of $\bar{\alpha} \geq \alpha - 2$ (see the "Methods" section). This condition covers important classes of long-range interactions such as van der Waals interactions ($\alpha = 6$) and dipole–dipole interactions ($\alpha = 3$). The condition $\alpha > 2$ is the most general sufficient condition for inequality (5) to be satisfied. Hence, when considering special classes of Hamiltonians, this condition can be relaxed. As one such example, we consider fermionic systems with long-range hopping as follows:

$$H = \sum_{i<j} \frac{1}{r_{i,j}^\alpha} (A_{i,j} a_i a_j^\dagger + B_{i,j} a_i a_j + \text{h.c.}) + V, \qquad (7)$$

where $\{a_i^\dagger, a_i\}_{i=1}^n$ are the creation and the annihilation operators for the fermion, and $V$ is composed of arbitrary finite-range interaction terms such as $a_i a_i^\dagger a_j a_j^\dagger$ with $r_{i,j} = \mathcal{O}(1)$. In the above cases, we can prove that for $\alpha > 3/2$, condition (5) is satisfied (see Lemma 2 in Supplementary Note 1). For $V = 0$, this model is integrable and exactly solvable. For example, the Kitaev chain with long-range hopping corresponds to this class. Interestingly, in the long-range Kitaev chain, the point $\alpha_c = 3/2$ is linked to a phase transition resulting from conformal-symmetry breaking[29].

Finally, we mention the relevance to experimental observations regarding the long-range area law. Recent advances in experimental setups have achieved direct observation of the second-order Rényi entropy[45]. The second-order Rényi entropy for a subsystem $L$ (as in Fig. 1) is defined as $S_2(\rho_L) := -\log[\mathrm{tr}(\rho_L^2)]$, and $S_2(\rho_L)$ provides a lower bound for the entanglement entropy $S(\rho_L)$ in Eq. (1). Hence, we can obtain the same area-law bound as (6) for $S_2(\rho_L)$. Recently, the measurement of Rényi entropy was reported[46] in long-range $XY$ models with tunable power exponents $0 < \alpha < 3$. We expect that our area-law bound would support the outcome of experimental observations regarding entanglement entropy of ground states.

**Matrix-product-state approximation.** Based on our analysis, we can also determine the efficiency of the approximation of ground states $|0\rangle$ in terms of the matrix-product representation. We approximate the exact ground state $|0\rangle$ using the following

quantum state $|\psi_D\rangle$:

$$|\psi_D\rangle = \sum_{s_1, s_2, \ldots, s_n=1}^{d} tr(A_1^{[s_1]} A_2^{[s_2]} \cdots A_n^{[s_n]})|s_1, s_2, \ldots, s_n\rangle,$$

where each of the matrices $\{A_i^{[s_i]}\}_{i,s_i}$ is described by the $D \times D$ matrix. We refer to the matrix size $D$ as the bond dimension. This MPS has entanglement entropy less than $\log D$ for an arbitrary cut of the system. Although arbitrary quantum states can be described by the MPS, generic quantum states require exponentially large bond dimensions, namely $D = \exp[\mathcal{O}(n)]$[18]. If a quantum state is well approximated by the MPS with small bond dimensions, we can efficiently calculate the expectation values of local observables (e.g., energy).

The MPS is the basic ansatz for various types of variational methods (e.g., the density-matrix renormalization group[18]), and it is crucial to determine whether ground states can be well approximated by the MPS with a small bond dimension. On the MPS representation of the ground state $|0\rangle$, we prove the following statement: if condition (5) is satisfied and the spectral gap is nonvanishing, there exists an MPS $|\psi_D\rangle$ with bond dimensions $D = \exp[c'\bar\alpha^{-1}\log^{5/2}(1/\delta)]$ [$c'$: constant, $\bar\alpha = \mathcal{O}(1)$] such that

$$\left\| tr_{X^c}(|\psi_D\rangle\langle\psi_D|) - tr_{X^c}(|0\rangle\langle0|) \right\|_1 \leq \delta|X| \qquad (8)$$

for an arbitrary concatenated subregion $X$, where $\| \cdot \|_1$ is the trace norm and $|X|$ denotes the cardinality of $X$. We show the proof in the "Methods" section.

From approximation (8), to achieve an approximation error of $\delta = 1/\mathrm{poly}(n)$, we need quasi-polynomial bond dimensions, namely $D = \exp[\mathcal{O}(\log^{5/2}(n))]$. Our result justifies the MPS ansatz with small bond dimensions, obtained at a moderate computational cost. This in turn explains the empirical success of the density-matrix-renormalization-group algorithm in long-range interacting systems[27,29,33]. On the other hand, our estimation is still slightly weaker than polynomial-size bond dimensions $D = \exp[\mathcal{O}(\log(n))]$. This is in contrast to the short-range interacting cases, where only sublinear bond dimensions $D = \exp[\mathcal{O}(\log^{3/4}(n))]$ are required to represent the gapped ground states using the MPS[6].

## Discussion

We discuss several future research directions and open questions. First, could we find an explicit example that violates the entanglement area law for $\alpha \leq 2$ or for $\alpha \leq 3/2$ in free fermionic systems? So far, rigorous violations of the area law have been observed for $\alpha = 1$ in gapped free fermionic systems[47]. Moreover, at $\alpha \approx 1$, all existing area-law violations are at most logarithmic, namely $S(\rho_L) \lesssim \log(|L|)$. The existence of a natural long-range interacting gapped system where the entanglement entropy obeys the subvolume law as $S(\rho_L) \lesssim |L|^\gamma \ (0 < \gamma < 1)$ is an intriguing issue. Conversely, it is also challenging to generalize our area law to the sub-volume-law bound for $\alpha \leq 2$. This regime is more relevant to high-dimensional systems, and any entropic bound better than the volume law would be helpful in tackling the high-dimensional area-law conjecture.

Second, can we develop an efficiency-guaranteed algorithm to calculate the ground state under the gap condition? In inequality (8), we have proved the existence of an efficient MPS description of the ground state, but how to find such a description is not clear. In short-range interacting systems, this problem has been extensively investigated in popular works by Vidick et al.[7,19]. We expect that their formalism would be generalized to the present cases, and leads to a quasi-polynomial-time algorithm for

calculating ground states within a polynomial error $1/\mathrm{poly}(n)$. Furthermore, we still have scope to improve the quasi-polynomial bond dimension of $\exp[\mathcal{O}(\log^{5/2}(n))]$ to approximate the ground states. Whether this bound can be relaxed to a polynomial form of $\exp[\mathcal{O}(\log(n))] = \mathrm{poly}(n)$ is a question that will be addressed in the future.

## Methods

**Derivation of $\bar\alpha \geq \alpha - 2$.** We here show the proof of $\bar\alpha \geq \alpha - 2$ for Hamiltonian (3). More general cases including fermionic systems are given in Supplementary Note 1. For the proof, we estimate the upper bound of

$$\| V_{X,Y} \| \leq \sum_{i \in X}\sum_{j \in Y} \| h_{i,j} \| \leq J\sum_{i \in X}\sum_{j \in Y} r_{i,j}^{-\alpha},$$

where we use the power-law decay of the interaction as $\| h_{i,j} \| \leq J/r_{i,j}^\alpha$. Let us define $\mathrm{dist}(X, Y) = r$. Then, we obtain

$$J\sum_{i \in X}\sum_{j \in Y} r_{i,j}^{-\alpha} \leq J\sum_{x=0}^{\infty}\sum_{y=0}^{\infty} (r + x + y)^{-\alpha},$$

where we use the fact that $X$ and $Y$ are concatenated subsets. For arbitrary integer $r_0 \in \mathbb{N}$, we have

$$\sum_{x=0}^{\infty}(x + r_0)^{-\alpha} \leq r_0^{-\alpha} + \int_{r_0}^{\infty} x^{-\alpha}dx \leq \frac{\alpha}{\alpha - 1}r_0^{-\alpha+1},$$

and hence

$$J\sum_{x=0}^{\infty}\sum_{y=0}^{\infty}(r + x + y)^{-\alpha} \leq \frac{\alpha J}{\alpha - 2}r^{-\alpha+2}.$$

We thus prove that $\|V_{X,Y}\|$ decays at least faster than $r^{-\alpha + 2}$.

**Proof sketch of the main result.** We here show the sketch of the proof for area-law inequality (6). The full proof is quite intricate, and we show the details in Supplementary Notes 2–4. In Fig. 3, we have summarized a flow of the discussions in this section.

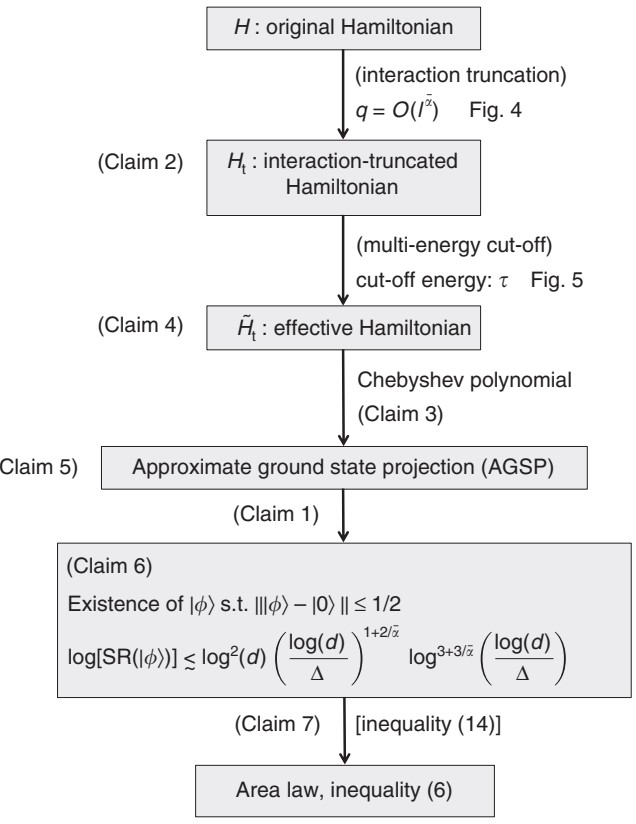

**Fig. 3 Flowchart of the area-law proof.** The proof consists of several key claims. The details of the proof for these claims are given in Supplementary Notes 2-4.

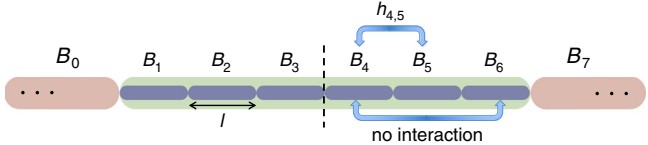

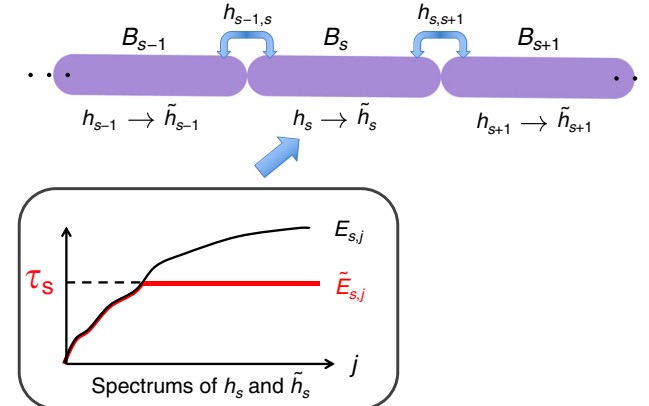

**Fig. 4 Interaction-truncated Hamiltonian $H_t$.** We truncate the long-range interactions only around the boundary. In the figure above, the interactions between nonadjacent blocks (i.e., $\{B_s\}_{s=0}^{7}$) are truncated. By this truncation, the properties of the Hamiltonian $H_t$ are proved to be almost the same as the original one $H$, as long as $ql^{-\bar{\alpha}} \lesssim 1$ (see Claim 2).

For the proof, we take the approximate-ground-state-projection (AGSP) approach[5,6]. The AGSP operator $K$ is roughly given by the operator that satisfies $K|0\rangle \simeq |0\rangle$ and $\| K(1 - |0\rangle\langle0|) \| \simeq 0$. The ground state $|0\rangle$ does not change by the AGSP $K$, while any excited state approximately vanishes by $K$. In more formal definitions, the AGSP is defined by three parameters $\delta_K$, $\epsilon_K$, and $D_K$. Let $|0_K\rangle$ be a quantum state that does not change by $K$, namely $K|0_K\rangle = |0_K\rangle$. Then, the three parameters are defined by the following three inequalities:

$$\| |0\rangle - |0_K\rangle \| \le \delta_K, \quad \| K(1 - |0_K\rangle\langle0_K|) \| \le \epsilon_K, \quad SR(K) \le D_K,$$

where $SR(K)$ is the Schmidt rank of $K$ with respect to the given partition $\Lambda = L \sqcup R$. The essential point of this approach is that a good AGSP ensures the existence of a quantum state that has a small Schmidt rank and large overlap with the ground state. It is mathematically formulated by the following statement:

## Claim 1

(Proposition 2 in Supplementary Note 2) Let $K$ be an AGSP operator for $|0\rangle$ with the parameters $(\delta_K, \epsilon_K, D_K)$. If we have $\epsilon_K^2 D_K \le (1/2)$, there exists a quantum state $|\psi\rangle$ with $SR(|\psi\rangle) \le D_K$ such that

$$\| |\psi\rangle - |0\rangle \| \le \epsilon_K \sqrt{2D_K} + \delta_K. \tag{9}$$

where $SR(|\psi\rangle)$ is the Schmidt rank of $|\psi\rangle$ with respect to the given partition.

From this statement, the primary problem reduces to one of finding a good AGSP to satisfy the condition $\epsilon_K^2 D_K \le (1/2)$.

In the construction of the AGSP operator with the desired properties, we usually utilize a polynomial of the Hamiltonian. The obstacle here is that the long-range interactions induce an infinitely large Schmidt rank in the thermodynamic limit; that is, the Hamiltonian $H$ has the Schmidt rank of poly($n$). In order to avoid this, we truncate the long-range interactions of the Hamiltonian. If we truncate all the long-range interactions, the norm difference between the original Hamiltonian and the truncated one is on the order of $\mathcal{O}(n)$, and hence the spectral gap condition cannot be preserved. The first central idea in the proof is to truncate the long-range interaction only around the boundary (see Fig. 4). In more detail, we first decompose the total system into $(q + 2)$ blocks with $q$ an even integer. The edge blocks $B_0$ and $B_{q+1}$ have arbitrary sizes, but the bulk blocks $B_1, B_2, \dots, B_q$ have the size $l$ (i.e., $|B_s| = l$). Then, we truncate all the interactions between nonadjacent blocks, which yields the Hamiltonian $H_t$ as

$$H_t = \sum_{s=0}^{q+1} h_s + \sum_{s=0}^{q} h_{s,s+1}, \tag{10}$$

where $h_s$ is the internal interaction in the block $B_s$, and $h_{s,s+1}$ is the interaction between two blocks $B_s$ and $B_{s+1}$. By using notation (4), we have $h_{s,s+1} = V_{B_s, B_{s+1}}$. In the Hamiltonian $H_t$, long-range interactions only around the boundary are truncated, and hence the norm difference between the original and the truncated Hamiltonian can be sufficiently small for large $l$.

## Claim 2

(Lemmas 3 and 4 in Supplementary Note 2) The norm distance between $H$ and $H_t$ is bounded from above by

$$\| H - H_t \| \le g_0 q l^{-\bar{\alpha}}.$$

Also, the spectral gap $\Delta_t$ of $H_t$ and the norm difference between $|0\rangle$ and $|0_t\rangle$ are upper-bounded by

$$\Delta_t \ge \Delta - 2g_0 q l^{-\bar{\alpha}}, \quad \| |0\rangle - |0_t\rangle \| \le \frac{\| H - H_t \|}{\Delta - 4\| H - H_t \|},$$

where $|0_t\rangle$ is the ground state of $H_t$.

From this statement, if $ql^{-\bar{\alpha}} \lesssim 1$, the truncated Hamiltonian $H_t$ possesses almost the same properties as the original one.

The second technical obstacle is the norm of the Hamiltonian. The gap condition provides us an efficient construction of the AGSP operator, which is expressed by the following statement:

## Claim 3

(Lemma 11 in Supplementary Note 2) By using the Chebyshev polynomial, we can find a $m$-degree polynomial $K(m, H_t)$ such that

$$\| K(m, H_t)(1 - |0_t\rangle\langle0_t|) \| \le 2\exp\left(-\frac{2m}{\sqrt{\| H_t \| / \Delta_t}}\right), \tag{11}$$

where the explicit form of the polynomial $K(m,x)$ is given in Supplemental Lemma 11.

We notice that the gap condition plays a crucial role in this claim. Here, the norm of $\|H_t\|$ is as large as $\mathcal{O}(n)$, which necessitates the polynomial degree of $m = \mathcal{O}(\sqrt{n})$. Polynomials with such a large degree cannot be utilized to prove the condition for the AGSP in Claim 1. To overcome this difficulty, we aim to construct an effective Hamiltonian with a small norm that retains the similar low-energy properties to the original Hamiltonian. For this purpose, in each of the blocks, we cut off the energy spectrum up to some truncation energy (see Fig. 5). Then, the block–block interactions (i.e., $h_{s,s+1}$) do not change, and the internal Hamiltonian $h_s$ is transformed to $\tilde{h}_s$. By this energy cutoff, the total norm of the effective Hamiltonian $\tilde{H}_t$ is roughly given by $q\tau$. The question is whether this effective Hamiltonian possesses the ground-state property similar to $H$. By extending the original result in ref. [48], which considers a cutoff in a Hamiltonian of one region, we prove the statement as follows:

## Claim 4

(Theorem 5 in Supplementary Note 2) Let us choose $\tau$ such that

$$\tau \gtrsim \log(q).$$

Then, the spectral gap $\tilde{\Delta}_t$ of the effective Hamiltonian is preserved as

$$\tilde{\Delta}_t \ge \mathcal{O}(\Delta_t).$$

Moreover, the norm distance between the original ground state $|0_t\rangle$ and the effective one $|\tilde{0}_t\rangle$ is exponentially small with respect to the cut-off energy $\tau$:

$$\| |\tilde{0}_t\rangle - |0_t\rangle \| \le e^{-\mathcal{O}(\tau)}.$$

As long as $\tau$ is larger than $\mathcal{O}(\log(q))$, the spectral gap is preserved, and the norm of the effective Hamiltonian is as large as $q\log(q)$, namely $\| \tilde{H}_t \| \lesssim q\log(q)$. In the standard construction of the effective Hamiltonian[6,48], we perform the energy cutoff only in the edge blocks (i.e., $B_0$ and $B_{q+1}$). However, this simple procedure allows us to prove the long-range area law only in the short-range power-exponent regimes (i.e., $\alpha > 3$). The multienergy cutoff is crucial to prove the area law even in the long-range power-exponent regimes (i.e., $2 < \alpha \le 3$).

By using the polynomial $K(m,x)$ in (11) with $x = \tilde{H}_t$, we can obtain the AGSP operator $K_t$ for the ground state $|0_t\rangle$ of $H_t$. Before showing the AGSP parameter for $K_t$, we discuss the Schmidt rank of the polynomial of the Hamiltonian. Now, the effective Hamiltonian $\tilde{H}_t$ is given by the form of $\sum_{s=0}^{q+1} \tilde{h}_s + \sum_{s=0}^{q} h_{s,s+1}$. By extending the Schmidt rank estimation in refs. [5,6], we can derive the following statement:

**Fig. 5 Effective Hamiltonian $\tilde{H}_t$ by multienergy cutoff.** In each of the internal Hamiltonians $\{h_s\}_{s=0}^{q+1}$, we perform the energy cutoff up to the energy $\tau_s = E_{s,0} + \tau$. Here, $\{E_{s,j}, |E_{s,j}\rangle\}$ are the energy eigenvalues and the corresponding eigenstates of $h_s$, respectively. The internal Hamiltonians $h_s$ and $\tilde{h}_s$ have the same eigenstates $\{|E_{s,j}\rangle\}$ and the same eigenvalues (i.e., $E_{s,j} = \tilde{E}_{s,j}$), as long as $E_{s,j} \le \tau_s$, above which the eigenvalues differ from each other.

## Claim 5

(Proposition 4 in Supplementary Note 2) The Schmidt rank of the power of the effective Hamiltonian $SR(H_t^m)$ is bounded from above by

$$SR(\tilde{H}_t^m) \le e^{\mathcal{O}(ql) + \mathcal{O}(m/q)\log(ql)}.$$

This inequality gives the upper bound of the Schmidt rank for $K(m, \tilde{H}_t)$.

We have obtained all the ingredients to estimate the parameters $\delta_{K_t}$, $\epsilon_{K_t}$, and $D_{K_t}$ for the AGSP $K_t = K(m, \tilde{H}_t)$. They are given by Claim 4, inequality (11), and Claim 5 as follows:

$$\delta_{K_t} = e^{-\mathcal{O}(\tau)}, \quad \epsilon_{K_t} = e^{-\mathcal{O}(m)/\sqrt{q\log(q)}},$$
$$\text{and} \quad D_{K_t} = e^{\mathcal{O}(ql) + \mathcal{O}(m/q)\log(ql)}, \tag{12}$$

where we omit the $\bar{\alpha}$-dependence of the parameters. Let us apply Claim 1 to the AGSP $K_t$ and the ground state $|0_t\rangle$. Under the condition of $ql^{-\bar{\alpha}} \lesssim 1$, we can find $q$, $m$, and $l$ such that $\epsilon_{K_t}^2 D_{K_t} \le (1/2)$, where $\{q, m, l\}$ have quantities of $\mathcal{O}(1)$. This leads to the following statement:

## Claim 6

(Proposition 6 in Supplementary Note 3) There exists a quantum state $|\phi\rangle$ such that $\| |0\rangle - |\phi\rangle \| \le 1/2$ with

$$\log[SR(|\phi\rangle)] \le c^* \log^2(d) \left(\frac{\log(d)}{\Delta}\right)^{1+2/\bar{\alpha}} \log^{3+3/\bar{\alpha}}\left(\frac{\log(d)}{\Delta}\right), \tag{13}$$

where $c^*$ is a constant that depends only on $g_0$ and $\bar{\alpha}$, which is finite in the limit of $\bar{\alpha} \to \infty$.

Finally, we construct a set of the AGSP operators $\{K_p\}_{p=1}^{\infty}$ for the ground state $|0\rangle$, where the AGSP parameters are denoted by $\delta_p$, $\epsilon_p$, and $D_p$. The errors $\epsilon_p$ and $\delta_p$ decrease with the index $p$, namely $\epsilon_1 \ge \epsilon_2 \ge \cdots$ and $\delta_1 \ge \delta_2 \ge \cdots$. In the limit of $p \to \infty$, the AGSP $K_p$ approaches the exact ground-state projection as $K_\infty = |0\rangle\langle 0|$, namely $\lim_{p\to\infty} \delta_p = 0$ and $\lim_{p\to\infty} \epsilon_p = 0$. These AGSP operators allow the derivation of an upper bound of the entanglement entropy, as well as the approximation of the ground state by quantum states with small Schmidt ranks.

## Claim 7

(Proposition 3 in Supplementary Note 2) Let $|\phi\rangle$ be an arbitrary quantum state with $\| |0\rangle - |\phi\rangle \| \le 1/2$. Also, let $\{K_p\}_{p=1}^{\infty}$ be AGSP operators defined as above. Then, we prove for each of $\{K_p\}_{p=1}^{\infty}$

$$\left\| \frac{K_p e^{-i\theta_p}|\phi\rangle}{\| K_p|\phi\rangle \|} - |0\rangle \right\| \le \gamma_p := \frac{\epsilon_p}{1/2 - \delta_p} + \delta_p,$$

where the phase $\theta_p \in \mathbb{R}$ is appropriately chosen. Moreover, under the condition $\gamma_p \le 1$ for all $p$, the entanglement entropy is bounded from above by

$$S(|0\rangle) \le \log[SR(|\phi\rangle)] - \sum_{p=0}^{\infty} \gamma_p^2 \log\frac{\gamma_p^2}{3D_{p+1}},$$

where we set $\gamma_0 := 1$.

In Proposition 7 of Supplementary Note 3, we show a construction of the AGSP set $\{K_p\}_{p=1}^{\infty}$ such that $\gamma_p^2 = 1/p^2$ and

$$\log(3D_p) \le c_1 \bar{\alpha}^{-1} \frac{\log^{5/2}(3p/\Delta)}{\sqrt{\Delta}} + c_2 \frac{\log^{3/2}(3p/\Delta)\log(d)}{\sqrt{\Delta}}, \tag{14}$$

where $c_1$ and $c_2$ are constants that depend on $g_0$. We have obtained the quantum state $|\phi\rangle$ with the Schmidt rank as in (13), and hence from Claim 7, the above AGSP operators give the upper bound of the entanglement entropy in (6). This completes the proof of the area law in long-range interacting systems. □

**MPS approximation of the ground state**. We here prove inequality (8). For simplicity, let us consider $X$ to be the total system (i.e., $X = \Lambda$). Generalization to $X \subset \Lambda$ is straightforward. Our proof relies on the following statement:

## Claim 8

(Lemma 1 in ref. [49]) Let $|\psi\rangle$ be an arbitrary quantum state. We define the Schmidt decomposition between the subsets $\{1, 2, \ldots, i\}$ and $\{i+1, i+2, \ldots, n\}$, as follows:

$$|\psi\rangle = \sum_{m=1}^{\infty} \mu_m^{(i)} |\psi_{\le i,m}\rangle \otimes |\psi_{>i,m}\rangle, \tag{15}$$

where $\{\mu_m^{(i)}\}_{m=1}^{\infty}$ are the Schmidt coefficients in the descending order. Then, there exists an MPS approximation $|\psi_D\rangle$ with the bond dimension $D$ that approximates the quantum state $|\psi\rangle$ as

$$\| |\psi\rangle - |\psi_D\rangle \|^2 \le 2\sum_{i=1}^{n-1} \delta_i, \quad \delta_i := \sum_{m>D} |\mu_m^{(i)}|^2.$$

From this claim, if we can obtain the truncation error of the Schmidt rank, we can also derive the approximation error by the MPS.

In the following, we give the truncation error by using Claim 7. Let us consider a fixed decomposition as $\Lambda = L \sqcup R$. Then, Claim 7 ensures the existence of the approximation of the ground state $|0\rangle$ with the approximation error $\gamma_p$, which is achieved by the quantum state $|\psi_p\rangle := K_p e^{i\theta_p}|\phi\rangle / \| K_p|\phi\rangle \|$ with its Schmidt rank of

$$\log[SR(|\psi_p\rangle)] \le \log(D_p) + \log[SR(|\phi\rangle)],$$

where $|\phi\rangle$ has the Schmidt rank of (13) at most. We have already proved that for $\gamma_p = 1/p^2$, the quantity $D_p$ is upper-bounded by (14). Thus, for $p \ge (1/\delta)^{1/4}$ or ($\gamma_p \le \delta^{1/2}$), the Schmidt rank $\log[SR(|\psi_p\rangle)]$ satisfies the following inequality:

$$\log[SR(|\psi_p\rangle)] \lesssim (\bar{\alpha}^{-1}\log(1/\delta) + 1)\log^{3/2}(1/\delta) \tag{16}$$

for $1/\Delta = \mathcal{O}(1)$ and sufficiently small $\delta \ll 1$, where we use the fact that $\log[SR(|\phi\rangle)]$ is a constant of $\mathcal{O}(1)$.

In order to connect inequality (16) to the truncation error of the Schmidt decomposition, we use the following statement:

## Claim 9

(Eckart–Young theorem[50]) Let us consider a normalized state $|\psi\rangle$ as in Eq. (15). Then, for an arbitrary quantum state $|\psi'\rangle$, we have the inequality of $\sum_{m>SR(|\psi'\rangle)} |\mu_m^{(i)}|^2 \le \| |\psi\rangle - |\psi'\rangle \|^2$, where the Schmidt rank $SR(|\psi'\rangle)$ is defined for the decomposition of $\{1, 2, \ldots, i\}$ and $\{i+1, i+2, \ldots, n\}$.

In the above claim, we choose $|0\rangle, |\psi_p\rangle$ as $|\psi\rangle, |\psi'\rangle$, respectively, and obtain the inequality of

$$\sum_{m>SR(|\psi_p\rangle)} |\mu_m^{(i)}|^2 \le \gamma_p^2, \tag{17}$$

where we use $\| |\psi_p\rangle - |0\rangle \| \le \gamma_p$. By applying inequalities (16) and (17) to Claim 8, we can achieve

$$\| |0\rangle - |\psi_D\rangle \|^2 \le 2n\delta,$$

if $\log(D)$ is as large as $\bar{\alpha}^{-1}\log^{5/2}(1/\delta)$ [$\bar{\alpha} = \mathcal{O}(1)$]. This completes the proof. □

## Data availability

Data sharing is not applicable to this paper, as no datasets were generated or analyzed during the current study.

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

## Acknowledgements
The work of T.K. was supported by the RIKEN Center for AIP and JSPS KAKENHI Grant no. 18K13475. K.S. was supported by JSPS Grants-in-Aid for Scientific Research (JP16H02211 and JP19H05603).

## Author contributions
T.K. and K.S. contributed to all aspects of this work.

## Competing interests
The authors declare no competing interests.
