## [Peer Review File · Nature Communications]

Reviewers' comments:

Reviewer #1 (Remarks to the Author):

Since the first proof of an entanglement area law for one-dimensional systems by Hastings (2007), many generalizations and improvements have appeared in the literature. Estimates of entanglement play an important role in many results in quantum information and proofs of better or more general estimates are therefore generally of considerable interest. The result reported in this manuscript could similarly be of great significance. Before we can evaluate the result, however, the authors will need to clarify the precise statement that is claimed and write a proof that can be reviewed for correctness. The reviewer encountered issues with both of these requirements in the first few pages of the short paper and the supplementary material.

Statement: apart from an unimportant change in the notation for the state, the entanglement entropy inequality (6) in the article and (S.14) in the supplemental material look the same. The role of the constant c in (6) is played by c_1 in (S.14), but the authors make different claims about the dependence of this constant on the parameters of the problem. c is said to depend on $\bar{\alpha}$ and g_0 while c_1 is claimed to depend only on k , α , and η . Which is the truth? Either way, there is a problem with the claims made following the statements of the inequality about the case $\alpha \rightarrow \infty$, and $\eta \rightarrow 0$. Both limits are taken as if c is independent of these parameters, contradicting the statements about them made just above.

Proofs: the proof of the main result is not written in good standard mathematical style. Be that as it may, perhaps this is not expected for Supplemental Material for articles in Nature Communications. However, already on the first two pages of the supplemental material some inequalities appear to be wrong:

- in (S.3) the local potentials present in the definition of the Hamiltonian in (3) appear to have disappeared. If they are present, the inequality (S.3) is wrong since one has extra terms on the LHS corresponding to $r=0$.

- inequality (S.4) for the gap Δ would only be correct if g defined in (S.3) is in fact given by the larger quantity appearing on the LHS of the equation. In any case the proof in Section I.1 of this very simple inequality is incomplete. The correct inequality is given in equation (1.1) of Commun. Math. Phys. 276, 437–472 (2007), DOI 10.1007/s00220-007-0342-z, where a simple proof is also given.

Perhaps the core argument of the authors is correct and the authors do indeed prove a new area law, but from the current manuscript this appears impossible to ascertain.

Reply to the referee

Since the first proof of an entanglement area law for one-dimensional systems by Hastings (2007), many generalizations and improvements have appeared in the literature. Estimates of entanglement play an important role in many results in quantum information and proofs of better or more general estimates are therefore generally of considerable interest. The result reported in this manuscript could similarly be of great significance. Before we can evaluate the result, however, the authors will need to clarify the precise statement that is claimed and write a proof that can be reviewed for correctness. The reviewer encountered issues with both of these requirements in the first few pages of the short paper and the supplementary material.

Reply: We would like to thank the referee for the careful reading of our manuscript. In the following, we show the point-by-point replies to the comments. We checked all the calculations of the draft again and revised it to improve the readability. The important revised parts are highlighted by the dark-red texts in the new manuscript.

Statement: apart from an unimportant change in the notation for the state, the entanglement entropy inequality (6) in the article and (S.14) in the supplemental material look the same. The role of the constant c in (6) is played by c_1 in (S.14), but the authors make different claims about the dependence of this constant on the parameters of the problem. c is said to depend on $\bar{\alpha}$ and g_0 while c_1 is claimed to depend only on k , α , and η . Which is the truth?

Reply: Thank you for the comment, which is very helpful to improve the readability of the draft. We revised the manuscript emphasizing the fundamental parameters in our theorems ^{*1}.

Below, we list the fundamental parameters in our theorems.

1. d : Dimension of the Hilbert space of one site,
2. Δ : Spectral gap between the ground state and the first excited state,
3. k : The number used in k -body interaction, i.e., the maximum number of sites involved in interactions (see Eq. (S.1) in the supplement),
4. g_0 : Defined in Assumption 1 in the supplement,
5. $\bar{\alpha}$: Defined in Assumption 1 in the supplement (Decay exponent of the block-block interactions).

^{*1}As we will show later, we have stopped using the parameter η .

In fact, as another fundamental parameter, we also use the following quantity:

$$\max_{i \in \Lambda} \sum_{Z: Z \ni i} \|h_Z\|, \quad (\text{R.1})$$

which has been defined by g in the supplement. In the previous version of the draft, the parameter g was used in different meanings between the main paper and the supplement, which caused confusions. In the present version, we use this parameter only for one meaning. In addition, the parameter g appears only in the supplement. In the previous draft, the main Hamiltonian (3) in the main text was

$$H = \sum_{i < j} h_{i,j} + \sum_{i=1}^n h_i \quad \text{with} \quad \|h_{i,j}\| \leq \frac{g}{r_{i,j}^\alpha}, \quad (\text{R.2})$$

which contained the parameter g . However, in the present draft, we changed it to

$$H = \sum_{i < j} h_{i,j} + \sum_{i=1}^n h_i \quad (\text{R.3})$$

with $\|h_{i,j}\| \leq J/r_{i,j}^\alpha$ and $\|h_i\| \leq B$ for $\forall i, j$, which does not contain the parameter g .

In the main text of the revised draft, for the sake of readability, we do not introduce the quantity in Eq. (R.1). Then, the coefficient c in the main text depends on $\{\alpha, J, B, k, g_0, \bar{\alpha}\}$ ^{*2}. In the supplement, we get into the detail of the calculation introducing the quantity in Eq. (R.1). Then, we set the quantity g to 1 without loss of generality by taking an appropriate unit. This makes the details of calculation easier. Hence, in the supplement, we describe the statement using $\{d, \Delta, k, g_0, \bar{\alpha}\}$, and the coefficient c_1 (defined as c_0 in the revised manuscript) depends only on $\{k, g_0, \bar{\alpha}\}$. In the present version of the supplement, we make a remark on this putting a footnote at the statement of theorem 1. We would like to emphasize that the parameter dependences of the coefficients have no inconsistencies between the main text and the supplement.

Either way, there is a problem with the claims made following the statements of the inequality about the case $\alpha \rightarrow \infty$, and $\eta \rightarrow 0$. Both limits are taken as if c is independent of these parameters, contradicting the statements about them made just above.

Reply: We realized that using η causes ambiguity and confusion. Thank you very much for this important comment. Therefore, we decided to give the main statement without using the parameter η . In the revised draft, the area law upper bound is explained as

$$S(|0\rangle) \leq c_0 \log^2(d) \left(\frac{\log(d)}{\Delta} \right)^{1+2/\bar{\alpha}} \log^{3+3/\bar{\alpha}} \left(\frac{\log(d)}{\Delta} \right). \quad (\text{R.4})$$

Also, we added the phrase “ c_0 is a constant which depends only on $k, g_0, \bar{\alpha}$, which has a finite value in the limit of $\bar{\alpha} \rightarrow \infty$.” We also mention this point in the proof.

^{*2}Moreover, we set a specific value $k = 2$ in the main text for simplicity.

In the revised version, the limit of $\bar{\alpha} \rightarrow \infty$ can be safely taken.

Proofs: However, already on the first two pages of the supplemental material some inequalities appear to be wrong: - in (S.3) the local potentials present in the definition of the Hamiltonian in (3) appear to have disappeared. If they are present, the inequality (S.3) is wrong since one has extra terms on the LHS corresponding to $r = 0$.

Reply: Thank you for the comment. In the previous draft, we have missed mentioning on the case of $r = 0$. In the revised version, we add a condition on the 1-local terms (i.e., on-site potentials) as follows:

$$\max_{i \in \Lambda} \|h_{\{i\}}\| \leq B. \quad (\text{R.5})$$

Then, we revise the inequality (S.3), which is now (S.5) in the revised manuscript, as in the following form:

$$\max_{i \in \Lambda} \sum_{Z: Z \ni i} \|h_Z\| = B + \max_{i \in \Lambda} \sum_{r=1}^{\infty} \sum_{Z: Z \ni i, \text{diam}(Z)=r} \|h_Z\| \leq B + \frac{\alpha J}{\alpha - 1} =: g. \quad (\text{R.6})$$

- inequality (S.4) for the gap Δ would only be correct if g defined in (S.3) is in fact given by the larger quantity appearing on the LHS of the equation. In any case the proof in Section I.1 of this very simple inequality is incomplete. The correct inequality is given in equation (1.1) of Commun. Math. Phys. 276, 437–472 (2007), DOI 10.1007/s00220-007-0342-z, where a simple proof is also given.

Reply: Thank you for showing the reference. This point is closely related to the definition of g . In this part, we have aimed to prove

$$\Delta \leq 2 \max_{i \in \Lambda} \sum_{Z: Z \ni i} \|h_Z\|, \quad (\text{R.7})$$

which has been also given in Commun. Math. Phys. 276, 437–472 (2007). In the revised version, the definition (R.6) of the parameter g correctly implies

$$\Delta \leq 2g. \quad (\text{R.8})$$

We refer to the reference [Commun. Math. Phys. 276, 437–472 (2007)] in the revised manuscript.

REVIEWERS' COMMENTS:

Reviewer #2 (Remarks to the Author):

This manuscript offers the statement of a fundamental result — whose proof is to be found in the very extensive Supplementary Information — namely the fact that ground states of one-dimensional gapped Hamiltonians with power-law interactions, decaying faster than $1/r^2$, and with interaction operators of finite norm, possess a ground state satisfying an area-law scaling of entanglement entropy. I think that the result per se is rather important - given the role that log-corrected area laws of entanglement play in the efficient encoding of quantum states using matrix-product states (MPS); and in the following I shall not object the correctness of its derivation. My main doubt about the appropriateness of the present material for a journal such as Nature Communications is the presentation style, and its readability for a broad audience.

Indeed, the present work is to be fully ascribed to the field of mathematical physics, and its most important contribution is a formal proof of the main result of the paper, which takes a large part of the 42 pages (!) of Supplementary Information accompanying the main text. On the contrary, the main text itself, beyond the statement of the proven bound for the entanglement entropy, is rather empty of original material. The long introductory section is dedicated to motivating the importance of models with power-law interactions in the context of recent theoretical studies as well as of quantum simulators. The original results are fully contained in the one-and-a-half page of the “Main results” section, which actually only contains the statement of the bound on the entanglement entropy, and of the scaling of the bond dimension for an MPS efficiently reproducing the ground state. Rather surprisingly there is no mention of how the proof of the bound on the entropy and on the MP bond dimension were obtained — not even a glimpse into the proof, which nonetheless represents the core of the original contribution offered by this work. There is no statement about why the long-range correlations, induced by the long-range interactions, play no role in altering the scaling of entanglement entropy with respect to short-range interactions: hence, as far as the main text go, we are encouraged to take the main result as the outcome of a mathematical tour-de-force, without any space for physical intuition. As a matter of fact, the main text is very little informative in general, and it contains a lot of material which seems to serve the only scope of “decorating” the main result without adding much useful information. For instance, out of the three figures of the main text, Fig. 1 and Fig. 3 bring essentially no information: Fig. 1 shows a bipartition of a chain, and Fig. 3 illustrates the structure of an MPS, something which can be found in dozens of papers on the subject; the only “useful” figure is Fig. 2, which in fact simply illustrates the parameter $\bar{\alpha}$ entering in the proven bound.

I am stressing all these aspects since the poverty of content of the main text is in stark contrast with the 42 pages of Supplementary Information, which represent in my view the “real” original work. They are a long sequence of theorems and lemmas which are rather technical and not particularly accessible to a large public.

The stylistic choice of summarizing the main results at the beginning of the paper, and offering the technical proofs later, is rather common in the mathematical physics literature, and it is perfectly justified when all the material (statement of the results plus proofs) are contained in one and the same paper. But here all technical aspects are relegated to the SI, and what remains in the main text is very little. In view of this I am very unsure whether this manuscript deserves publication in Nature Communications in the present format. It may in my opinion, if the authors rewrite substantially the main text, by cutting all the useless material (such as the figures I pointed at) and provide instead an accessible discussion of the main points of their proof of the bound, as well as some physical intuition behind it. If this is not possible, I suspect that the material at hand is not appropriate for the broad readership of Nature Communications, and it is really better fit for a specialized journal in mathematical physics.

I end my report with some more localized remarks:

1) the proof of the bound on the entanglement entropy appears to apply only for interactions decaying faster than $1/r^2$, as well as for degrees of freedom with a finite local Hilbert space, but this crucial statement is completely missing from the abstract. For instance the proof would not

apply to models of lattice bosons with e.g. dipolar interactions — something which is of great relevance to magnetic atoms or dipolar molecules — because bosonic operators have infinite norm. I think that these limitations are very important and they should be declared in the abstract right away;

2) the authors mention that Eq. 2 (the transverse-field Ising model) can be realized by trapped ions with interactions decaying as $1/r^\alpha$ where α is contained between 0 and 3; they could mention as well that the very same model can be implemented with Rydberg atoms, in that case with $\alpha = 6$ (van-der-Waals interactions);

3) in the caption of Fig. 2 the authors suggest that the bound $g_0 r^{-\bar{\alpha}}$ on the norm of the interaction operator for two extended regions X and Y at a distance r *implies* that they can only consider $\alpha > 2$. I confess that this implication is far from obvious to me;

4) it is unclear to me how one goes from the bound on the entropy to the bound on the bond dimension (above Eq. 8), and I am not sure to what extent the proof is given in the SI; in particular the bond dimension D does not depend at all on the $\bar{\alpha}$ exponent appearing in the bound on the entropy, something which is rather surprising at first sight.

Reviewer #3 (Remarks to the Author):

REPORT ON NCOMMS-19-39336A

DATE: APRIL 20, 2020

AUTHOR(S): TOMOTAKA KUWAHARA, KEIJI SAITO

AREA LAW OF NON-CRITICAL GROUND STATES IN 1D LONG-
TITLE: RANGE INTERACTING SYSTEMS

RECEIVED: 2019-10-10 03:34:36.0

Key results. Entanglement properties of ground states have a history of providing key insights into their physical properties and more importantly about their efficient approximations. Tensor networks (matrix product states in 1d and projected entangled pair states in 2d+) are known to efficiently approximate ground states of local Hamiltonians. The authors extend previous results: As the authors explain, their parameter α (describing the fall off of interactions) implies that their model falls into the universality class of short-range interactions ($\alpha = \infty$) for $\alpha \geq 3$. The authors show that in general for $\alpha \geq 2$ and in certain situations for even smaller α (namely free fermions with $\alpha \geq 3/2$), the ground state entanglement entropy satisfies an area law and that the ground state can be efficiently approximated by a matrix product state (i.e. with appropriate scaling of the bond dimension). This result is new and it challenges previous work by Koffel et. al., where a sub-logarithmic violation was indicated by numerical results. I believe it is worthy of publication.

Originality and significance. The result provides a clear criterion, when the ground state of a gapped long-range interacting system satisfies an area law and can be efficiently approximated by a matrix product state. Unfortunately, the authors cannot give an explicit construction of such a state, but knowing both area law and that it can be approximated efficiently is already a meaningful progress that is relevant for the study of such systems, which includes various models that can even be experimentally studied. The authors showed a high level of originality, effort and determination by modifying the existing proof technique of using approximate ground-state projection (AGSP). It appears that a crucial step in the presented proof is the construction of the effective Hamiltonian, where the energy cut-off is not only

performed on the edge blocks, but in each block. This provides the crucial improvement by showing the result for general situations with $\alpha < 3$.

Validity. The manuscript consists of the main text, which is merely a 3-page summary of the result with limited discussion and a 42-page supplement providing a detailed proof, which itself is split into 10 propositions, 17 lemmata and 5 theorems. A large part of the proof techniques is based on applying and relating different inequalities on the operators norms, the norm of the state difference and the scaling of the entanglement entropy. While a large part of this consists of tedious estimates, the authors gave the overall proof of the main theorem enough structure to identify and understand the individual building blocks. Both the proof as a whole and several individual pieces (such as using the multi-energy cut-off in each of the blocks) showed some original ideas. The revisions have improved the presentation and readability of the proof.

Presentation. The overall structure of the paper works with a clear statement of the central result in the main text, while a detailed proof with a comprehensible outline and even a flow chart of the proof is attached. I would have liked to see a short and more compact version of the proof, but believe that this may have obscured some parts and trust that the authors made a reasonable effort to keep the presentation readable. The use of figures and simple illustrations was helpful. Overall, the writing is clear, even though there are occasional unusual choices of words or less common grammatical structures. I will indicate some examples below, but overall I did not feel that this would obstruct the understanding of the derivations. I believe that it would be useful to include another table listing all symbols/conventions that are used in more than one proof etc. of the supplement. The table could list each symbol, a brief explanation to it and where it is introduced/defined/used in the supplement. This would have helped me to have a quick reference (since when looking for the definition of a symbol, one only needs to look in the section that one is reading and in the table) and the authors already did this for certain variables that their final estimate depends on (see table I). I liked the very compact outlook sections, which lays out a number of interesting avenues and natural follow up questions resulting from the present manuscript.

Individual corrections. The following bullet points contain a few suggestions regarding on rephrasing and indication of some minor typos.

- **Main text**

- Some sentences are missing words like ‘the’ or ‘a’ (for example: “ However, providing detailed proof of the area law”). These are always minor issues.

- After equation (2), the authors could elaborate a bit on the implications of the universality classes being the same for $\alpha > 3$ regarding the entanglement scaling.
- Just two examples of phrases that the authors may want to rephrase slightly: “a challenging problem both in numerical and theoretical levels” and “Using the small-incremental-entangling theorem [36], this assumption allows us to prove the area law in generic gapped short-range interacting systems.” (here, I first thought the authors are referring to their own work in the present paper or in reference [26]).

• **Supplement**

- In (S.2) and later, they use $\sum_{Z:\text{condition}}$. I was not familiar with this notation, but figured it just means to sum over all Z satisfying the condition.
- g_0 and $\bar{\alpha}$ in table I are introduced in (S.10), but you refer to (S.8).
- The main text argues that condition 5 is always satisfied for $\alpha > 2$ because $\bar{\alpha} \geq \alpha - 2$. It would be good to refer explicitly, where this relation is shown (supplement III?).
- In supplement III, Lemma 1 shows $\bar{\alpha} = 2 - \alpha$ under certain conditions and lemma 2 shows $\bar{\alpha} = 3/2 - \alpha$ under other conditions. It seems that there is a flip of sign compared to the main text, where $\bar{\alpha} \geq \alpha - 2$ is claimed.
- Below (S.40), you assume that K is the Hermite operator. I assume you mean “ K is a Hermitian operator”?
- The second sentence after (S.40) is a bit weird. Please rephrase.
- I noticed that the authors use for certain symbols bars, rather than introducing new variables (for example $\bar{\alpha}$). In many situations, such variables are related in the sense that they play a similar role in different parts (like scaling exponent), but I wasn’t sure what the systematics was (later they also use $\bar{\gamma}$). For the ground state approximation, they use $|\bar{0}\rangle$, but also $|\tilde{0}\rangle$ (when AGSP approximation). It all works, but maybe the authors could mention their logic to make it easier to remember, while reading (just an idea).

Reply to Reviewer 2

This manuscript offers the statement of a fundamental result — whose proof is to be found in the very extensive Supplementary Information — namely the fact that ground states of one-dimensional gapped Hamiltonians with power-law interactions, decaying faster than $1/r^2$, and with interaction operators of finite norm, possess a ground state satisfying an area-law scaling of entanglement entropy. I think that the result per se is rather important - given the role that log-corrected area laws of entanglement play in the efficient encoding of quantum states using matrix-product states (MPS); and in the following I shall not object the correctness of its derivation. My main doubt about the appropriateness of the present material for a journal such as Nature Communications is the presentation style, and its readability for a broad audience.

Indeed, the present work is to be fully ascribed to the field of mathematical physics, and its most important contribution is a formal proof of the main result of the paper, which takes a large part of the 42 pages (!) of Supplementary Information accompanying the main text. On the contrary, the main text itself, beyond the statement of the proven bound for the entanglement entropy, is rather empty of original material. The long introductory section is dedicated to motivating the importance of models with power-law interactions in the context of recent theoretical studies as well as of quantum simulators. The original results are fully contained in the one-and-a-half page of the “Main results” section, which actually only contains the statement of the bound on the entanglement entropy, and of the scaling of the bond dimension for an MPS efficiently reproducing the ground state. Rather surprisingly there is no mention of how the proof of the bound on the entropy and on the MPS bond dimension were obtained – not even a glimpse into the proof, which nonetheless represents the core of the original contribution offered by this work.

Reply: We would like to appreciate the referee for careful reading and the quite positive assessment of this work. All the comments are quite helpful for us to improve the main part of the paper. Following your suggestions, we have substantially rewritten the manuscript in the main part. In the following, please let us address the additional comments one by one. The revised parts are highlighted by the dark-red texts in the new manuscript.

There is no statement about why the long-range correlations, induced by the long-range interactions, play no role in altering the scaling of entanglement entropy with respect to short-range interactions: hence, as far as the main text go, we are encouraged to take the main result as the outcome of a mathematical tour-de-force, without any space for physical intuition.

Reply: Thank you for the comment. We have added one section to explain the physical intuition. Naive explanation is the following: Even under the long-range interactions, the bi-partite correlation function still decays polynomially (Hastings and Koma, CMP, 2006), and hence if the

correlation decay is sufficiently fast, the area law seems to be satisfied. This intuition indicates that for $\alpha \gg 1$ the area law is expected to hold. In fact, this intuition is not correct, since there exists a quantum state with super-polynomially decaying correlations which breaks the area law (Hastings, QIC, 2016).

The points are raised as follows:

1. The gap condition is much stronger than decay of the bi-partite correlations. In other words, under the assumption of the gap, there are various additional constraints on the entanglement structure in ground states (Kuwahara et al., QST, 2017). This gap condition is reflected by the efficient approximation of the ground state projection by using a polynomial of the Hamiltonian.
2. The condition $\alpha > 2$ is deeply related to the condition that the boundary interaction is finite. Even for thermal state at a finite temperature, we have the same condition for the area law. It is natural to expect that the condition for the area law in the thermal state should be looser than that in the ground state. We thus expect that the condition $\alpha > 2$ is necessary for the long-range area law at zero temperature.

We have summarized these two points in Section of “**Why does the area law holds for $\alpha > 2$.**” We note that these discussion is not rigorous, and hence there is still room to improve the condition of $\alpha > 2$. We have shown this point in the section of Discussion.

As a matter of fact, the main text is very little informative in general, and it contains a lot of material which seems to serve the only scope of “decorating” the main result without adding much useful information. For instance, out of the three figures of the main text, Fig. 1 and Fig. 3 bring essentially no information: Fig. 1 shows a bipartition of a chain, and Fig. 3 illustrates the structure of an MPS, something which can be found in dozens of papers on the subject; the only “useful” figure is Fig. 2, which in fact simply illustrates the parameter $\bar{\alpha}$ entering in the proven bound.

Reply: Thank you for your comment! We agree that Fig. 3 is unnecessary and hence we removed it. We added several pictures to understand the proof. We have remained Fig. 1 since it will be

still useful to understand the setup visually.

I am stressing all these aspects since the poverty of content of the main text is in stark contrast with the 42 pages of Supplementary Information, which represent in my view the “real” original work, They are a long sequence of theorems and lemmas which are rather technical and not particularly accessible to a large public. The stylistic choice of summarizing the main results at the beginning of the paper, and offering the technical proofs later, is rather common in the mathematical physics literature, and it is perfectly justified when all the material (statement of the results plus proofs) are contained in one and the same paper. But here all technical aspects are relegated to the SI, and what remains in the main text is very little. In view of this I am very unsure whether this manuscript deserves publication in Nature Communications in the present format. It may in my opinion, if the authors rewrite substantially the main text, by cutting all the useless material (such as the figures I pointed at) and provide instead an accessible discussion of the main points of their proof of the bound, as well as some physical intuition behind it. If this is not possible, I suspect that the material at hand is not appropriate for the broad readership of Nature Communications, and it is really better fit for a specialized journal in mathematical physics.

Reply: Thank you for your comment! We add several new sections to answer these points. After showing our main statement on the area law, we give a section to mention the physical intuition to obtain the condition $\alpha > 2$. Also, in Method section, we show several essential ideas to prove the area law. This section comprises of several Claims. If these Claims are assumed without proofs, non-specialists can follow the proof of the area law.

1) the proof of the bound on the entanglement entropy appears to apply only for interactions decaying faster than $1/r^2$, as well as for degrees of freedom with a finite local Hilbert space, but this crucial statement is completely missing from the abstract. For instance the proof would not apply to models of lattice bosons with e.g. dipolar interactions — something which is of great relevance to magnetic atoms or dipolar molecules — because bosonic operators have infinite norm. I think that these limitations are very important and they should be declared in the abstract right away;

Reply: Thank you for your comment! That’s right. We utilize the fact that one-site energy is bounded from by a finite constant. If we assume that the number of boson in one site is finitely bounded, we can use the same analyses. We added one phrase on this point in the abstract. Also,

in the main text, after the definition of the Hamiltonian, we again emphasize this point.

2) the authors mention that Eq. 2 (the transverse-field Ising model) can be realized by trapped ions with interactions decaying as $1/r^\alpha$ where α is contained between 0 and 3; they could mention as well that the very same model can be implemented with Rydberg atoms, in that case with $\alpha = 6$ (van-der-Waals interactions);

Reply: Thank you for your comment! We have added the case of $\alpha = 6$ with some references.

3) in the caption of Fig. 2 the authors suggest that the bound $g_0 r^{-\bar{\alpha}}$ on the norm of the interaction operator for two extended regions X and Y at a distance r *implies* that they can only consider $\alpha > 2$. I confess that this implication is far from obvious to me;

Reply: Thank you for your comment! We have given the derivation of the lower bound of $\bar{\alpha}$ in Lemmas 1 in the supplementary materials. As has been pointed out by Reviewer 3, this derivation should be given in the main text. So, we have added the proof for the case of two-body interactions in Method section.

4) it is unclear to me how one goes from the bound on the entropy to the bound on the bond dimension (above Eq. 8), and I am not sure to what extent the proof is given in the SI; in particular the bond dimension D does not depend at all on the $\bar{\alpha}$ exponent appearing in the bound on the entropy, something which is rather surprising at first sight.

Reply: Thank you for your comment! In fact, the MPS approximation cannot be obtained only from the area law for the entanglement entropy. The proof is based on the approximation error by the truncation of the Schmidt coefficient, which is included in the main theorem in Supplementary materials. In order to make this point clear, we added the proof in Method section. We also mention the $\bar{\alpha}$ dependence of the bond dimension D .

Reply to Reviewer 3

Key results. Entanglement properties of ground states have a history of providing key insights into their physical properties and more importantly about their efficient approximations. Tensor networks (matrix product states in 1d and projected entangled pair states in 2d+) are known to efficiently approximate ground states of local Hamiltonians. The authors extend previous results: As the authors explain, their parameter α (describing the fall off of interactions) implies that their model falls into the universality class of short-range interactions ($\alpha = \infty$) for $\alpha \geq 3$. The authors show that in general for $\alpha \geq 2$ and in certain situations for even smaller α (namely free fermions with $\alpha \geq 3/2$), the ground state entanglement entropy satisfies an area law and that the ground state can be efficiently approximated by a matrix product state (i.e. with appropriate scaling of the bond dimension). This result is new and it challenges previous work by Koffel et. al., where a sub-logarithmic violation was indicated by numerical results. I believe it is worthy of publication.

Originality and significance. The result provides a clear criterion, when the ground state of a gapped long-range interacting system satisfies an area law and can be efficiently approximated by a matrix product state. Unfortunately, the authors cannot give an explicit construction of such a state, but knowing both area law and that it can be approximated efficiently is already a meaningful progress that is relevant for the study of such systems, which includes various models that can even be experimentally studied. The authors showed a high level of originality, effort and determination by modifying the existing proof technique of using approximate ground-state projection (AGSP). It appears that a crucial step in the presented proof is the construction of the effective Hamiltonian, where the energy cut-off is not only performed on the edge blocks, but in each block. This provides the crucial improvement by showing the result for general situations with $\alpha < 3$.

Validity. The manuscript consists of the main text, which is merely a 3-page summary of the result with limited discussion and a 42-page supplement providing a detailed proof, which itself is split into 10 propositions, 17 lemmata and 5 theorems. A large part of the proof techniques is based on applying and relating different inequalities on the operators norms, the norm of the state difference and the scaling of the entanglement entropy. While a large part of this consists of tedious estimates, the authors gave the overall proof of the main theorem enough structure to identify and understand the individual building blocks. Both the proof as a whole and several individual pieces (such as using the multi-energy cut-off in each of the blocks) showed some original ideas. The revisions have improved the presentation and readability of the proof.

Reply: We would like to thank the referee for the positive assessment of the potential significance of our work. In the following, we show the point-by-point revisions to answer the remarks. The

revised parts are highlighted by the dark-red texts in the new manuscript.

Presentation. The overall structure of the paper works with a clear statement of the central result in the main text, while a detailed proof with a comprehensible outline and even a flow chart of the proof is attached. I would have liked to see a short and more compact version of the proof, but believe that this may have obscured some parts and trust that the authors made a reasonable effort to keep the presentation readable. The use of figures and simple illustrations was helpful Overall, the writing is clear, even though their are occasional unusual choices of words or less common grammatical structures. I will indicate some examples below, but overall I did not feel that this would obstruct the understanding of the derivations. I believe that it would be useful to include another table listing all symbols/conventions that are used in more than one proof etc. of the supplement. The table could list each symbol, a brief explanation to it and where it is introduced/defined/used in the supplement. This would have helped me to have a quick reference (since when looking for the definition of a symbol, one only needs to look in the section that one is reading and in the table) and the authors already did this for certain variables that their final estimate depends on (see table I). I liked the very compact outlook sections, which lays out a number of interesting avenues and natural follow up questions resulting from the present manuscript.

Reply: Thank you for your comment! In order that non-specialist can follow the essence of the proof, we have added Method section which describes the outline of the proof. The outline comprises of several claims without proofs (but, showing the corresponding Lemma/Proposition in Supplemental information). By combining them altogether, the area law can be derived with minimal effort.

Also, in the last section of the supplementary materials, we added a list of definitions and notations which we use several times in the proof.

Individual corrections. The following bullet points contain a few suggestions regarding on rephrasing and indication of some minor typos.

- **Main text**

- Some sentences are missing words like ‘the’ or ‘a’ (for example: “ However, providing detailed proof of the area law”). These are always minor issues.

Reply: Thank you for your comment! We commissioned a professional proofreading service for

our paper again. We believe that the new version is more readable.

– After equation (2), the authors could elaborate a bit on the implications of the universality classes being the same for $\alpha > 3$ regarding the entanglement scaling.

Reply: Thank you for your comment! We here would like to argue that $\alpha > 3$ is a kind of trivial regime and $\alpha \leq 3$ is more important. We add one sentence after “For $\alpha > 3$, the universality class is the same as that of short-range interacting systems [32,33] (i.e., $\alpha = \infty$).” as follows: “It means that the regime of $\alpha \leq 3$ is essentially important in discussing the area law in long-range interacting systems.”

– Just two examples of phrases that the authors may want to rephrase slightly: “a challenging problem both in numerical and theoretical levels” and “Using the small-incremental-entangling theorem [36], this assumption allows us to prove the area law in generic gapped short-range interacting systems.” (here, I first thought the authors are referring to their own work in the present paper or in reference [26]).

Reply: Thank you for your comment! We have rephrased them as “a challenging problem both in numerical and theoretical levels” \rightarrow “a challenging problem both numerically and theoretically.”
“Using the small-incremental-entangling theorem [36], this assumption allows us to prove the area law in generic gapped short-range interacting systems.” \rightarrow “Using the small-incremental-entangling theorem, this assumption ensures the area law in generic gapped short-range interacting systems [36].”

● **Supplement**

– In (S.2) and later, they use $\sum_{Z:\text{condition}}$. I was not familiar with this notation, but figured it just means to sum over all Z satisfying the condition.

Reply: Thank you for your comment! We added the explanation of this notation in the set up section as follows: “we often use the notation of $\sum_{Z:\text{condition}}$ which means the summation over all Z satisfying the condition.”

– g_0 and $\bar{\alpha}$ in table I are introduced in (S.10), but you refer to (S.8).

Reply: Thank you for your comment! We corrected the typo.

– The main text argues that condition 5 is always satisfied for $\alpha > 2$ because $\bar{\alpha} \geq \alpha - 2$. It would be good to refer explicitly, where this relation is shown (supplement III?).

Reply: Thank you for your comment! We have added Method section to show the lower bound of $\bar{\alpha}$, derivation of the MPS approximation, and the sketch of the proof of the area law.

– In supplement III, Lemma 1 shows $\bar{\alpha} \geq \alpha - 2$ under certain conditions and lemma 2 shows $\bar{\alpha} \geq 3/2 - \alpha$ under other conditions. It seems that there is a flip of sign compared to the main text, where $\bar{\alpha} \geq \alpha - 2$ is claimed.

Reply: Thank you for your comment! We have corrected the typo.

– Below (S.40), you assume that K is the Hermite operator. I assume you mean "K is a Hermitian operator"?

Reply: Thank you for your comment! We have corrected it to "Hermitian operator."

– The second sentence after (S.40) is a bit weird. Please rephrase.

Reply: Thank you for your comment! We have rephrased the sentences as follows:

In the following, we characterize the approximate ground state projection (AGSP) operators by three parameters $\{\delta_K, \epsilon_K, D_K\}$. Let $|0_K\rangle$ be a quantum state that is invariant by K such that

$$K |0_K\rangle = |0_K\rangle. \tag{R.1}$$

Then, the parameters are defined by the following inequalities:

$$\| |0\rangle - |0_K\rangle \| \leq \delta_K, \quad \|K(1 - |0_K\rangle\langle 0_K|)\| \leq \epsilon_K, \quad \text{and} \quad \text{SR}(K) \leq D_K. \tag{R.2}$$

– I noticed that the authors use bars for certain symbols, rather than introducing new variables (for example $\bar{\alpha}$). In many situations, such variables are related in the sense that they play a similar role in different parts (like scaling exponent), but I wasn't sure what the systematics was (later they also use $\bar{\gamma}$). For the ground state approximation, they use $|\bar{0}\rangle$, but also $|\tilde{0}\rangle$ (when AGSP approximation). It all works, but maybe the authors could mention their logic to make it easier to remember, while reading (just an idea).

Reply: Thank you for your comment! We redefined the several notations and stopped using $\bar{\gamma}$ and $|\bar{0}\rangle$. Now, the $\bar{}$ sign is used only for $\bar{\alpha}$ and the $\tilde{}$ sign is used only for parameters which are relevant to the effective Hamiltonian. Also, we denote the approximate ground state by the AGSP as $|0_K\rangle$. All the new notations are summarized in the last section of the supplementary materials.